# Hemodynamic Analysis of the Geometric Features of Side Holes Based on GDK Catheter

**DOI:** 10.3390/jfb13040236

**Published:** 2022-11-11

**Authors:** Yang Yang, Yijing Li, Chen Liu, Jingyuan Zhou, Tao Li, Yan Xiong, Ling Zhang

**Affiliations:** 1College of Mechanical Engineering, Sichuan University, Chengdu 610065, China; 2Department of Nephrology, Kidney Research Institute, West China Hospital of Sichuan University, Chengdu 610041, China; 3Department of Applied Mechanics, Sichuan University, Chengdu 610065, China

**Keywords:** hemodynamic, geometric features, side holes, catheter, shear stress

## Abstract

Hemodialysis is an important means to maintain life in patients with end-stage renal disease (ESRD). Approximately 76.8% of patients who begin hemodialysis do so through catheters, which play vital roles in the delivery of hemodialysis to patients. During the past decade, the materials, structures, and surface-coating technologies of catheters have constantly been evolving to ameliorate catheter-related problems, such as recirculation, thrombosis, catheter-related infections, and malfunction. In this study, based on the commercial GDK catheter, six catheter models (GDK, GDK1, GDK2, GDK3, GDK4, and GDK5) with different lumen diameters and different geometric features of side holes were established, and computational flow dynamics (CFD) were used to measure flow rate, shear stress, residence time (RT), and platelet lysis index (PLI). These six catheters were then printed with polycarbonate PC using 3D printing technology to verify recirculation rates. The results indicated that: (1) the catheter with a 5.5 mm outer diameter had the smallest average shear stress in the arterial lumen and the smallest proportion of areas with shear stress > 10 pa. With increasing catheter diameter, the shear stress in the tip volume became lower, the average RT increased, and the PLI decreased due to larger changes in shear stress; (2) the catheters with oval-shaped side holes had smaller shear stress levels than those with circular-shaped holes, indicating that the oval design was more effective; (3) the catheter with parallel dual side holes had uniformly distributed flow around side holes and exhibited lower recirculation rates in both forward and reverse connections, while linear multi-side holes had higher shear stress levels due to the large differences in flow around side holes. The selection of the material and the optimization of the side holes of catheters have significant impacts on hemodynamic performances and reduce the probability of thrombosis, thus improving the efficiency of dialysis. This study would provide some guidance for optimizing catheter structures and help toward the commercialization of more efficient HD catheters.

## 1. Introduction

More than 1 million patients die each year worldwide due to ESRD (end-stage renal disease), and up to 1.7 million patients with acute kidney injury die due to lack of access to effective treatment, 85% of which deaths occur in developing countries [1]. Hemodialysis is one of the kidney replacement methods for the treatment of ESRD [2]. For dialysis treatment, building an ideal vascular pathway is the primary preparation before treatment, and it is also a necessary condition to successfully achieve the treatment effect [3]. In the United States, approximately 80% of patients began hemodialysis with a catheter in 2011 [4]. In China, more than 100,000 hemodialysis patients need to have a double-lumen catheter fitted at least once (temporary or long-term) for dialysis treatment [5].

Despite hemodialysis (HD) catheters being widely used and despite the low cost of their placement and replacement, they have certain defects, such as being associated with high rates of thrombosis, infection, and dysfunction [6]. Catheter materials have significant impacts on the prevention of catheter-related infection and the life of hemodialysis catheters. The ideal catheter material would be biocompatible, hemocompatible, biostable, chemically neutral, stable, and deformable in accordance with environmental forces [7]. Traditional biomaterials include polyurethane and silicone. Through advances in materials technology, there has been a transition to the use of polyurethane or carbon ethane (polyurethane/polycarbonate copolymer), which have better catheter strength and flexibility while maintaining larger internal diameters. The properties of the material used determine the mechanical properties of a catheter to some extent, while the surface coating on the catheter material is related to the biocompatibility and anticoagulation properties of the catheter. In addition, the design of catheters is also closely related to blood recirculation and thrombus formation [8].

In recent years, studies on catheter design have investigated the design of the lumen, the distal tip, and the side holes, along with hemodynamic performance by means of CFD, animal, or in vitro experiments [9]. For example, Ling et al. [10] compared the clinical and rheologic outcomes of HD catheters with step, split, or symmetrical tips in patients; Vesely et al. [6] found that symmetric-tip and step-tip designs had the advantage of more stable fluid flow patterns and lower recirculation rates compared to split-tip designs in a model simulating hemodialysis treatment; Ogawa et al. [11] observed and evaluated the recirculation and fluid characteristics of the catheter tips for three catheters with different tip shapes; Tal et al. [12] compared catheters with or without side holes by performing an analysis of flow rates, infection rates, and survival rates; Owen et al. [13] investigated the impact of different side-hole configurations on a symmetrical-tip catheter by evaluating the local hemodynamics and catheter performance through CFD. The above studies focused on hemodynamic analyses of the existing products and models, and only a few studies have put forward new ideas about catheter design to improve hemodynamic performance. For instance, Cho et al. [8] put forward three new HD catheter designs and compared the effects of the catheters’ side holes and distal tips on hemodynamic factors with those of existing catheters, using CFD and in vitro methods, while Clark et al. [14] compared a new dialysis catheter (VectorFlow) with a Palindrome catheter in terms of shear stress, RT, PLI, and recirculation rate using the CFD method and a bench model of hemodialysis.

In the present study, the comparative analysis of new designs for catheter lumens and patterns of catheter side holes was the research focus. Through structural improvements based on the commercial GDK catheter (Gambro, Stockholm, Sweden), five newly designed catheter models are put forward, the CFD method and in vitro experiments having been used to study the effects of design parameters. The problems affecting catheter performance include platelet activation [15], flow stagnation regions [16], and the recirculation of dialyzed blood [17]. Platelet activation and aggregation are related to elevated regions of shear stress and prolonged blood transit time [18]. The higher the values of shear stress, PLI and RT lead to higher probabilities of thrombosis. Lower recirculation of dialyzed blood affects dialysis efficiency. Therefore, the variations in lumen diameter and the different geometric features of the side holes were assessed among these six types of catheters by comparing the values for flow rate, shear stress, RT, and PLI of inflowing blood at the tip of catheter along with the recirculation rates of these 3D-printed catheters to spot possible design issues.

This study could help achieve catheter designs with more optimized structures, longer life, and higher dialysis efficiencies. The results could ultimately lead to better-performing catheters to improve the life cycle of patients with kidney disease.

## 2. Methods

Six catheter models were simulated by the finite element method through three-dimensional modeling, meshing, and numerical simulation, and their hemodynamic parameters, such as flow rate, shear stress, RT, and PLI, were obtained. Then, these catheter models were 3D-printed with polycarbonate PC, and their recirculation rates were tested by dye tracing in in vitro experiments.

### 2.1. Geometry of Hemodialysis Catheters

The GDK catheter, as shown in Figure 1, has a symmetric design; the inner diameter is 2.6 mm and the outer is 3.6 mm, and the thickness of the board which separates the arterial and venous lumens is 0.5 mm. The three circular side holes with a vertical distribution and the distal tips with nozzle shapes are located on the venous lumen. Five circular side holes for the inflow are located on the venous lumen. All these circular side holes’ diameters are 2 mm.

Keeping other structural elements unchanged, the outer diameters of the GDK catheter were expanded from 3.6 mm to 4.3 mm (GDK1) and 5.5 mm (GDK2, 3, 4, and 5), as shown in Figure 2, so the inner diameter was changed to 3.3 mm (GDK1) and 4.5 mm (GDK2, 3, 4, and 5). For GDK1 and GDK2, the distributions of side holes were kept the same as those for the GDK catheter to investigate the effects of the internal cavity on the flow pattern. In GDK3, the shape of the side holes was changed to oval, but the hole area was kept the same to compare the effects of the shapes of the side holes. For GDK4, the area of the side holes was enlarged from 3.14 mm^2^ to 6.28 mm^2^ to observe the effect of the area of the side holes on the flow. The side holes of the venous lumen were named side-hole1, side-hole2, and side-hole3, according to the distance from the tip. For GDK5, the distribution of the side holes was changed from linear to a parallel dual-hole pattern.

In the model domain, the 3D model of the superior vena cava (SVC) and the catheter is shown in Figure 3. The SVC was considered to be a cylinder with a diameter of 20 mm. In order to eliminate the impact of unrelated variables on performance, the lengths of the SVC and each catheter were set to 340 mm and 200 mm, respectively, and the distance between the distal tip of each catheter and the SVC was fixed to 100 mm so that the blood inflows to the catheters were fully normalized and no outlet effects occurred.

In the SVC fluid domain, a cylindrical cavity with a length of 20 mm and the same diameter as the outer diameter of the catheter was dug out at the front end of the root catheter, thus imposing a boundary condition. In order to better compare inflow parameters, a cuboid ‘tip volume’ was defined at the tip of the venous lumen for each catheter. Its width and height were equal to the outer diameter and radius of each catheter, respectively, such that it included the entire inflow lumen. In order to focus on the flow characteristics of the different catheters, the tip volume extended from the most distal point of the catheter up to 50 mm, where the flow velocity would become stable and fully developed and there would be no further flow interference.

All geometries were created using Solidworks software (Dassault System SolidWorks, Concord, MA, USA) and exported into Fluent Meshing software (Fluent Inc., Lebanon, NH, USA).

### 2.2. Mesh of Hemodialysis Catheters

The grid consisted of a mixture of tetrahedral and hexahedral grids, the number of which depended on the geometry of the catheter, and an encrypted grid at the ‘tip volume’ and side holes. In order to obtain the best calculation time, the grid was verified independently. Table 1 shows the maximum velocity and the average shear stress at the tip volume of the GDK with different numbers of grids, resulting in a grid consisting of 320,187 cells, which were mainly tetrahedral, and the inner and outer lumens of the catheters were distributed to 6 boundary layers. A cross-section of the grid is shown in Figure 4.

### 2.3. Governing Equations and Boundary Conditions

The fluid relevant here is blood, which is identified as an incompressible, uniform, non-Newtonian fluid [19], the viscosity of which varies with the shear rate. In previous studies, incompressible Newtonian fluid models were established as the working fluids; this study considered the change in shear rate occurring during inflow and used an asymptotic shear-thinning Carreau model [20], which was defined using the shear rate of Equation (1) and defined by Equation (2).
(1)$$\dot{\gamma }=\sqrt{2\times D_{ij}\cdot D_{ij}}$$
where $\dot{\gamma }\;$ is the shear rate and *D* is the strain rate tensor, with *i*, *j* = 1, 2, 3 as the inner projects.
(2)$$\mu =\mu_{\infty }+\left(\mu_{\infty }+\mu_{0}\right){\left[1+{\left(\lambda \dot{\gamma }\right)}^{2}\right]}^{\frac{n-1}{2}}.$$
where μ is the viscosity of blood and $\mu_{\infty }$ = 0.0345 Pa.s, *n* = 0.25, $\mu_{0}$ = 0.025 Pa.s, and λ = 25 s.

Boundary conditions: The SVC inlet was set to a 0.3 m/s velocity inlet, and the outlet was a pressure outlet, the gauge pressure of which was zero. The surfaces of the catheter and SVC were set as the wall without slip condition, and the inlet and outlet of the catheter were set as a mass flow inlet and mass flow outlet of 400 mL/min, respectively [21].

The platelet lysis index (PLI) (Equation (3)) was calculated to evaluate the possible damage occurring to the platelets. This index was first applied to the heart valve prostheses [22] and is widely used to evaluate the risk of platelet-activated aggregation and thrombosis in previous studies [14].
(3)$$PLI=3.66\cdot 10^{-6}\cdot tp^{0.77}\cdot \tau p^{3.075}.$$
where *t_p_* is the residence time of the platelet and *τ_p_* is the shear stress acting on the platelet. For the GDK catheter, side holes away from the tip were taken as the calculation path.

Each path flowed into the Poiseuillean flow zone from the side hole of the catheter and through the disturbance zone. The shear stress, velocity, and retention time were outputted at each point the path passed through, and the PLI was calculated at each step (0.5 μm).

Solution settings: The ANSYS Fluent COUPLED solver was used to solve the fluid numerically. The flow in the catheter was similar to that in other catheter studies [23,24], assuming laminar flow. The mass continuity residual magnitude was less than 10^−6^, and the combined flow rate of the tip and the side hole was equal to 400 ± 1 mL/min, these values being used to assess convergence.

### 2.4. In Vitro Experiment for Recirculation

The recirculation test bench for the dye tracing experiment is shown in Figure 5. The device simulates the superior vena cava mainly by a plexiglass tube (diameter 20 mm, length 500 mm). A steady flow is maintained in the plexiglass tube by connecting it to a peristaltic pump at the upper and lower adapters. The catheter gland can be adjusted to ensure that catheters with different external diameters are inserted into the plexiglass tube. The catheter gland is connected vertically to the catheter with a silicone rubber seal to achieve a complete seal inside the vessel.

According to decreasing order of thrombus formation, the catheter materials used were polyvinyl chloride, polyethylene, polyurethane, and silica gel. Silicone or polyethylene carbamate catheters are to be preferred because of their high smoothness, strong adhesion to resist fibers and pathogens, good histocompatibility, small vascular stimulation, lower chance of thrombosis, and reduced chances of infection and intravascular injury [25]. The six catheters were printed with polycarbonate PC material using 3D printing technology to verify the recirculation rates. A 3D print of the GDK catheter is shown in Figure 6. Polycarbonate PC is widely used in artificial kidney hemodialysis equipment and other medical equipment that needs to be operated under transparent, intuitive conditions and requires repeated sterilization.

The overall flow rate of the device is controlled by three dual-channel peristaltic pumps, as shown in Figure 7. The Rombauer BT100 peristaltic pump is connected to the arteriovenous lumen with a maximum flow rate of 570 mL/min, pumping fluid from Reservoir 3 into the glass column at a rate of 400 mL/min through the venous lumen of the hemodialysis catheter. Fluid is drawn from the glass column through the arterial lumen at 400 mL/min and then outputted to Reservoir 4. The BT-CA JIHPUMP BT-600CA peristaltic pump was used to maintain fluid flow through the glass column at 2400 mL/min to simulate the flow of blood from the superior vena cava. In practice, surgeons would resolve a catheter malfunction by reversing the direction of blood flow in the arterial and venous lumens [26]. In the experiment, the peristaltic direction of BT-100 was reversed, and the fluid from Reservoir 4 passed through the arterial lumen into the plexiglass tube at a flow rate of 400 mL/min, while the fluid from Reservoir 3 was withdrawn from the venous lumen at a flow rate of 400 mL/min to realize the reverse connection of the dialysis catheter.

Recirculation in the catheter means that when the dialyzed blood passes through the venous lumen and returns to the body’s superior vena cava to be withdrawn again by the arterial lumen the blood gets dialyzed again, which reduces the efficiency of hemodialysis [17].

Taking the forward connection as an example, we first ran the BT600-CA peristaltic pump to maintain a flow rate of 2400 mL/min in the Plexiglass tube, then simultaneously started the BT100 peristaltic pump to inject the dye liquid at a concentration of 1% into Reservoir 3, which flowed into the venous lumen. After a one-minute test, 50 mL samples were extracted from Reservoir 4. When the absorbance coefficient and the optical path of the dye solution are unchanged, the absorbance of the dye solution is proportional to the concentration of the dye solution [27], and the concentration of red dye in the fluid in the arterial lumen was measured using a UV–Visible spectrophotometer. After the completion of each experiment, the pump channel was connected to a peristaltic pump and pure water was used to discharge the dye reagents in the pump channel so as to avoid the impact of residual dye reagents and errors in subsequent experiments. Each experiment was repeated five times, and the average values and standard errors of the means (SEMs) were calculated to improve the accuracy of the experimental data. The recirculation rate (RR) values were calculated as shown in Equation (4).
(4)$$RR(\%)=\frac{Q_{a}\times C_{a}}{Q_{v}\times C_{v}}\times 100$$
where *Q_a_* and *Q_v_* are the flow rates of the arterial lumen and the venous lumen in the catheter, respectively, and *C_v_* and *C_a_* are the concentrations of the dye from the venous lumen and arterial lumen, respectively. In order to determine statistical significance, the non-parametric Kruskal–Wallis test was used, with *P* < 0.05, to compare the recirculation of the different catheters.

## 3. Results

### 3.1. Analysis of Flow Rate

Blood, after dialysis, flows out from the distal tip and the side holes in the venous lumen. The function of the side holes is to reduce flow velocity at the distal tip of the catheter (Q_tip_), leading to a lower recirculation rate.

The flow rates of the side holes and the distal tip in the venous lumen of the GDK, GDK1, GDK2, and GDK3 catheters are shown in Table 2, which also presents the percentage of flow rates through side holes/distal tips based on total flow for these catheters.

The flow rate of the side holes decreased with the distance of the side holes from the tip. In the case of the GDK catheter, the flow rate of side-hole1 near the distal tip was 89.71 mL/min, while those of side-hole2 and 3, away from the distal tip, were 38.74 and 14.57 mL/min, respectively. Similar trends were also seen for other catheters (GDK1, GDK2, and GDK3). When the outer diameter of the catheter was increased from 3.6 mm (GDK) to 4.3 mm (GDK1), the flow rate of the distal tip decreased from 257.14 mL/min to 173.98 mL/min, and the total flow of these side holes (Q_side_) increased from 143.02 mL/min to 225.78 mL/min. The Q_tip_’s proportion of total flow decreased from 64.28% to 43.49%, thereby increasing the cross-sectional areas of the inner cavity, which proved to be effective for improving the flow distribution in the catheter.

When the outer diameter was 5.5 mm, the Q_tip_ in GDK2 increased to 47.8%. When the side holes were changed from being circular-shaped with an area of 3.14 mm^2^ (GDK2) to oval-shaped with an area of 6.28 mm^2^ (GDK3), the flow rate of the distal tip decreased from 191.2 mL/min to 139.1 mL/min, and the Q_tip_’s proportion of total flow decreased from 47.8% to 34.77%. The flow rate of these side holes (Q_side_) increased from 208.8 mL/min to 260.9 mL/min. This indicates that the flow from the distal tip in the venous lumen can be varied by modifying the shape and area of the side holes, changing the flow through it. This suggests that larger outside diameters and oval side holes with larger areas on the venous lumen are all effective ways to improve the flow configuration in the catheter.

Although the arrangements are similar for GDK2 and GDK3, there are large differences in the flow distributions of each side hole. The flow rates for GDK2 with circular-shaped side holes along the distance to the tip were 188.1 mL/min, 109.71 mL/min, 61.7 mL/min, 27.65 mL/min, and 12.11 mL/min, respectively, from the far to near ends; the first two side holes took 74.4% of dialysis blood flow inlet in the arterial lumen. For GDK3, the values of flow rate for each side hole were changed to 194.5 mL/min, 112.1 mL/min, 59.66 mL/min, 24.85 mL/min, and 8.8 mL/min, respectively, because of the changes in the shape of the side holes. The hole flow distribution of the GDK4 catheter was changed further by the increased lateral hole area; the flow rate of two oval-shaped side holes away from the tip reached 95.4% of the total inlet flow. When the arrangement of the side holes was changed to parallel dual holes, both side holes would split inlet flow, as shown in Figure 8d; the flow rates for each side were about 200 mL/min in the arterial lumen.

The catheters with parallel side holes had lower flow rates than that with linear multi-side holes because of the flow being split evenly across each hole. It can be concluded that the structure of the side holes has an important effect on the flow distribution of the catheter.

### 3.2. Analysis of Shear Stress

The flow of blood through the catheter is considered laminar [28]. It is believed that shear stresses over 10 pa would cause damage to platelets and lead to thrombosis [29]. So, the maximum flow rate, the average shear stress, and the proportion of the area of shear stress over 10 Pa were determined from the tip volume of each of the six catheters (as shown in Table 3) to quantitatively assess hemodynamic performance.

It is shown that the percentage of shear stress regions > 10 Pa was 22.6% and that the maximum flow velocity was 5.14 m/s while the average shear stress was 13.6 pa at the tip volume of the commercial GDK catheter. When the outer diameter of the GDK catheter was enlarged from 3.6 mm (GDK) to 4.3 mm (GDK1), the percentage of regions where the shear stress at the tip volume exceeded 10 Pa in the arterial lumen increased to 27.4%, and the average shear stress decreased to 11 pa, with little change in the shear stress level.

When the outer diameter was 5.5 mm (GDK2), the shear stress at the tip volume in the arterial lumen changed significantly, with average shear stress decreasing from 11 pa to 3.43 pa, and the shear stress region > 10 pa also decreased to 12.8%. It was presumed that increasing the outer diameter would reduce the shear stress in the tip volume of the arterial lumen. The catheters (GDK2, GDK3, GDK4, and GDK5) with 5.5 mm outer diameters were all associated with similar levels of mean shear stress and percentage shear stress > 10 Pa.

When the area of the side holes in the arterial lumen was kept the same and the shape was changed from circular to oval, the shear stress level in the tip volume was changed, and the average shear stress in GDK3 with oval-shaped side holes was a little smaller than that in GDK2 with circular-shaped holes (3.15 pa < 3.43 pa), while, when the shape of the oval-shaped side holes was kept unchanged and the area was enlarged from 3.14 mm^2^ (GDK3) to 6.28 mm^2^ (GDK4), the average shear stress became slightly larger (3.15 pa < 3.19 pa) and the percentage of shear stress > 10 Pa at the tip volume for these catheters maintained similar levels (12.8%, 12.2%, and 12.6%). This indicates that when considering the variation in the shape and area of side holes acting on the level of shear stress at the tip volume of the catheter arterial lumen, oval-shaped side holes with smaller areas have a positive impact on reducing platelet activation as well as the risk of blood damage.

The GDK5 catheter at the tip volume was associated with lower average shear stress (3.16 Pa) compared with that of the GDK4 catheter, at 3.19 Pa. Since the flow rate of each side hole was so different, the average shear stress of the GDK4 catheter with linear multi-side holes was a little higher than that with the parallel double side-hole structure, which also reduced the maximum velocity at the tip volume from 1.78 m/s to 1.72 m/s.

### 3.3. Analysis of PLI

PLI, as a weighting of platelet models experiencing high shear stress and residence time, can reflect the activation state of platelets to assess the merit of catheter tip design. The values for the average residence time (RT), PLI, average shear stress, and percentage of shear stress >10 Pa at the tip volume of the six catheters are presented in Table 4.

Shear stress and retention time need to be comprehensively considered to analyze the effects of lumen diameter on catheter function. The PLI of the GDK2 catheter with an outer diameter of 5.5 mm is 3.5% of that of the catheter with an outer diameter of 3.6 mm (GDK); it can be observed that the outer diameter is positively correlated with the average residence time of the platelet model, while it is negatively correlated with the mean shear stress and PLI values.

In contrast to the circular-shaped side holes, the PLI value for the oval-shaped side holes was reduced from 0.0326 (GDK2) to 0.0253 (GDK3) and the average shear stress was also reduced from 5.867 Pa to 5.81 Pa. The changes in these values indicated that the change of side-hole shape would slightly reduce the average shear stress acting on the platelets and PLI, but reduce average residence time in the tip volume in these two catheters.

As shown in Table 4, low PLI was seen with the GDK4 (PLI = 0.0174) and GDK5 (PLI = 0.0124) catheters. GDK5 with parallel dual-side holes experienced slightly longer residence times (0.0409) than the design with linear multi-side holes but had a slightly lower PLI because of a small reduction in high shear stress regions > 10 Pa. It is also clear that the arterial lumen with parallel dual oval-shaped side holes is the most reasonable design and has the lowest PLI and average shear stress values.

Although a large amount of blood is delivered to the downstream region of the lumen, there is still blood circulating in the lumen due to an insufficient pressure gradient, which contributes to the maximum retention time, and the blood circulating in the lumen is also a trigger for thrombosis. The platelet inflows for the GDK2, GDK3, GDK4, and GDK5 catheters are visualized in Figure 6.

It can be seen from Figure 9a,b, with the same arrangement of side holes and side-hole areas, that the flow characteristics of the GDK3 catheter are better than those of GDK2 and that the average RT of the catheter with oval-shaped side holes is lower than that with circular-shaped holes (0.0253 m/s < 0.0326 m/s). This is due to the fact that, with the linear arrangement of side holes, the inflow from the second side hole is obstructed as it flows down the lumen by the inflow from the first side hole. When the shape of the side holes was changed from circular to oval, the blocking effect was reduced, and the flow from the second side hole entered the lumen downstream more quickly, so that, regarding the shape of the side holes in the arterial lumen, oval-shaped side holes have the effect of reducing the RT of blood compared to circular-shaped ones and reduce the risk of thrombosis. As shown in Figure 8b,c, because the areas of the side holes in the GDK4 catheter are larger than those in the GDK3, the pressure gradient in the GDK4 is not sufficient to allow all the blood to enter the downstream region of the lumen, resulting in the circulation of blood at the entrance to the catheter upstream of the lumen, thus making the average RT of the GDK4 catheter larger than that of the GDK3 catheter (0.0308 m/s > 0.0179 m/s). The GDK5 catheter with a parallel dual side-hole arrangement had theoretically lower average RT values than the linear side-hole design, as the inflow was split equally, allowing undisturbed inflow from both holes. However, because of the obvious area of blood self-circulation upstream of the arterial lumen, as can be seen in Figure 9d, which makes the blood turbulent and stagnant in the lumen, the average RT of the GDK5 catheter was larger than that of the GDK4 (0.0409 m/s > 0.0308 m/s).

The recirculation rates for the GDK catheter after structural changes were obtained by a dye tracing experiment. The catheter was connected forward and backward to evaluate the change in catheter dialysis efficiency with different connection methods. The reverse connection is the reversal of the direction of blood flow in the arteriovenous lumen in order to eliminate a thrombus. Each experiment was repeated five times, and the averages and standard errors of the mean values (SEM) were calculated to improve the accuracy of the experimental data. The results of the dye tracing experiment, as shown in Figure 10, indicated that the RRs of GDK, GDK1, and GDK2, which worked with forward and reverse usage, were 0.68 ± 0.28%, 4.69 ± 1.41%, 0.72 ± 0.24%, 5.09 ± 1.23%, 0.46 ± 0.26%, and 4.2 ± 1.14%, respectively; those of the dye tracing experiments showed that the values of RR for GDK3, GDK4, and GDK5, which worked with forward and reverse usage, were 0.26 ± 0.14%, 3.23 ± 0.96%, 0.86 ± 0.27%, 6.64% ± 1.27%, 0.33% ± 0.12%, and 2.86 ± 0.50%, respectively. The recirculation rates of the catheters with the reverse connection between the forward connection were statistically significant (*p* < 0.05, Kruskal–Wallis test). The recirculation rates of GDK5 with the reverse connection were statistically significant with respect to all catheters except GDK3 with the reverse connection. The method of connection and the distribution of the side holes play important roles in recirculation in catheters.

As can be seen from the experiment, all six catheters showed that the recirculation rate of the reverse connection was higher than that of the forward connection. The GDK3 catheter had the lowest recirculation rate for the forward connection, the GDK5 catheter had the lowest recirculation rate for the reverse connection, the GDK5 arterial lumen with double side holes had a lower recirculation rate compared to the other ones, and the GDK4 catheter exhibited the highest recirculation rate for the forward and reverse connections.

## 4. Discussion

With the continuous development of science and technology, new technologies, new materials, and new structures have been introduced, and new developments in materials, surface coatings, and structures have been introduced in clinical treatment. Catheter material is an important determinant in the prevention of catheter-related infection [7] and usually consists of high polymers (usually polyurethane or silicone). Catheter structural considerations mainly relate to the structure of the lumen, side hole, and tip.

In this study, we took the GDK catheter as a prototype, which has a symmetric tip with the same length as the arterial and venous lumens [9]. The flow characteristics of five types of new HDs, designated based on the GDK catheter were evaluated with CFD and in vitro experiments to investigate the effects of the outer diameter of the catheter, the geometry, and the arrangement of side holes on catheter hemodynamics. The results showed that the catheter with an outside diameter of 5.5 mm and parallel double oval-shaped side holes achieved relatively low shear stress levels and recirculation rates. These results would help catheter structural optimization design in the future, which aims to decrease the risk of dialysis treatment interruption caused by catheter-related infections and insufficient flow and reduce the economic burden and increase the life cycle of patients with kidney disease.

For the commercial GDK catheter, dialysis blood mainly flows out from the distal tip, up to 64.8% of total flow, and the flow rate of the side holes could not be kept high enough to prevent recirculation. Increasing the flow of side holes and decreasing the flow of distal tips helps to reduce the recirculation rate of the catheter and improve dialysis efficiency in symmetric-tip catheters [8]. In the study, two methods were put forward to optimize catheter structure: one is to enlarge the lumen of the catheter, and the other is to change the geometric features of the side holes. The results showed that when the outside diameter of the GDK catheter was enlarged to 5.5 mm and the side-hole shape was oval with a larger area a greater outlet flow through the side holes and a high flow-rate region located in the tip volume would achieve a lower rate of catheter recirculation. The experiments also verified that the GDK3 catheter connection had the lowest recirculation rate. These findings are consistent with those of Cho et al. (2021), who found that larger side holes and a nozzle-shaped distal tip could reduce the flow rate and high shear stress region and improve catheter effectiveness, while Mareels et al. [15] found that the side holes helped reduce shear stress and RT. Shear stress as a risk factor for platelet activation leads to thrombosis. A threshold of 10 Pa of shear stress for platelet activation has been identified, and areas of flow stagnation or recirculation can induce platelet aggregation. It is necessary for HD catheters to reduce blood recirculation and shear stress to ensure performance [6]. For the arterial lumen, it was found that as the outer diameter of the catheter increases, the flow rate of dialysis blood within the catheter decreases, the blood RT increases, the shear stress level in the tip volume decreases, and the PLI decreases further. Therefore, for the design of a catheter, although a larger outer diameter may cause a certain degree of increase in blood RT, due to the reduced level of shear stress in the arterial lumen, increasing the outer diameter would help to reduce the probability of thrombosis and thus increase catheter life.

Side hole design has been a controversial issue. The number and size of side holes play an important role in blood recirculation [24]. Multiple side holes are used in the arterial lumen design for the GDK catheter so that when the side holes of the arterial lumen are blocked by a generated thrombus during the dialysis process, the remaining side holes can be used as a backup entrance to continue the dialysis process. However, in the study, the flow rate of the side holes increased, as the side holes were positioned away from the tip, and the two side holes far from the tip took up as much as 74.4% of the flow rate. As shown in Figure 8, the side holes far from the distal tip were more associated with increased turbulence and subsequent intra-luminal thrombosis [30], so these side holes cannot be used as backup inlets to ensure the normal operation of dialysis after thrombus formation. Therefore, this structure has to be optimized to parallel dual side holes (GDK5) to keep each side hole in the arterial lumen serving as a means of blood suction. It was found that both side holes were assumed to be taken as the blood inlet end, and the flow was evenly divided, which indicated that the parallel dual side holes served to equalize the flow; when one side hole was clogged by a thrombus, the other side hole could continue to provide suction to reduce catheter malfunction. In this study, changing the side holes in the arterial lumen from circular to oval further improved the hemodynamics of the catheter. The large size of the side holes was advantageous, such that GDK4, with oval-shaped side holes, had a better performance, and the oval-shaped side holes as means of inflow allowed for a slow flow of blood into the catheter, indicating that these oval-shaped side holes are more suitable for long-term use compared to circular-shaped ones.

Since the side holes are mostly boreholes with rough walls, the more side holes there are, the more thrombi are generated at the surface of the side holes, which block the catheter in the inner lumen, leading to lower blood performance and shorter dialysis lifespan [31]. The GDK catheter is based on the idea that when blood flow is blocked by thrombus generation in the side holes, more side holes represent more alternate entrances to reduce catheter power loss. In contrast, the design of parallel side holes in the arterial lumen is superior to the original GDK catheter design because blood is evenly distributed and flows do not affect each other, such that dialysis function is not affected when one side hole is blocked due to thrombosis.

Blood recirculation after dialysis is prone to occur in the presence of reverse connections, thus affecting catheter dialysis efficiency. In this paper, the recirculation rates of different catheters were measured by dye tracing experiments. The variation of recirculation depends on the area, geometry, and arrangement of side holes. Most recirculation in dialysis catheters occurs at the distal tip, so it is necessary to reduce the flow rate at the distal tip and increase the proportion of flow through the side holes to reduce recirculation. Through dye tracing experiments, we found that the area of side holes has a certain effect on the recirculation rate and that the GDK4 catheter has a higher forward and reverse recirculation rate than the other catheters, which is due to the fact that the blood after dialysis does not enter into the main circulation but flows from the venous lumen and enters again from the arterial lumen and the purified blood is dialyzed twice, thus reducing the efficiency of hemodialysis.

The arrangement of the side holes has an effect on the recirculation rate of the catheter, and in the experiment catheters with parallel dual side-hole arrangements exhibited lower recirculation rates in both forward and reverse connections. Therefore, in the future commercialization of dialysis catheters, the parallel dual side-hole structure can be a new direction for structural optimization. These improvements would improve hemodialysis efficiency and thus reduce the economic and physical burden of patients with chronic renal failure.

There are several limitations to the study. The first is the that the study only focused on a single manufacturer’s catheter; the design and analysis of the other five new catheter models were based on commercial GDK catheters. Another limitation of this study is that in vitro experiments do not allow for the complete evaluation of catheter function, such that experiments with animals should be required in future studies.

## 5. Conclusions

Optimal catheter design is an urgent need for patients with nephropathy and its ultimate goal is to reduce the failure of dialysis treatment due to unreasonable catheter structure. In this study, we analyzed the design of HD catheters, including changes to diameters and side holes based on the GDK catheter, in order to evaluate their effects on hemodynamic factors, such as flow rate, shear stress, PLI, RT, and RR, through numerical simulation and in vitro experiments. The results indicated that larger outer diameters and oval-shaped side holes can reduce average shear stress in the arterial lumen and that oval-shaped side holes are effective for reducing RRs. The parallel dual side-hole structures have uniformly distributed side hole flow and the combination of a 5.5 mm diameter and parallel dual oval shapes for side holes exhibited better hemodynamic properties, thus providing better performance than the existing models.

## Figures and Tables

**Figure 1 jfb-13-00236-f001:**
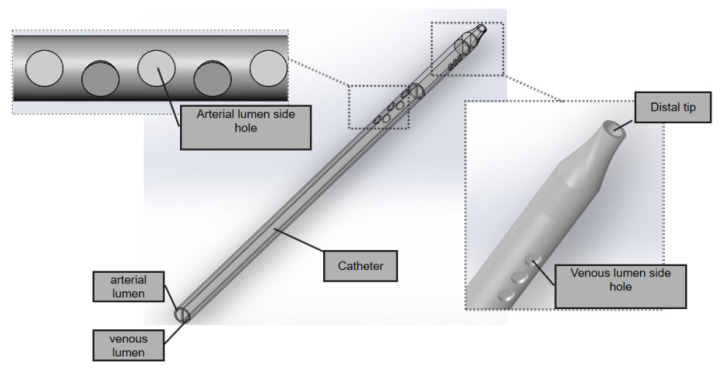
The GDK catheter.

**Figure 2 jfb-13-00236-f002:**
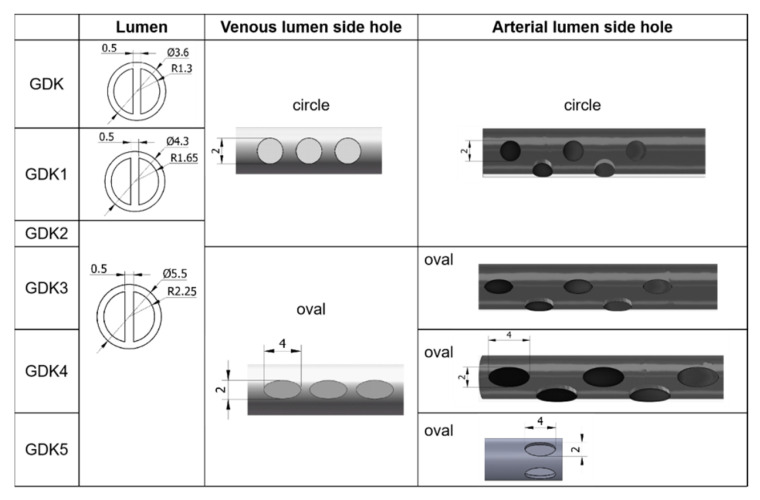
The side holes in the lumens of different configurations of GDK catheters.

**Figure 3 jfb-13-00236-f003:**
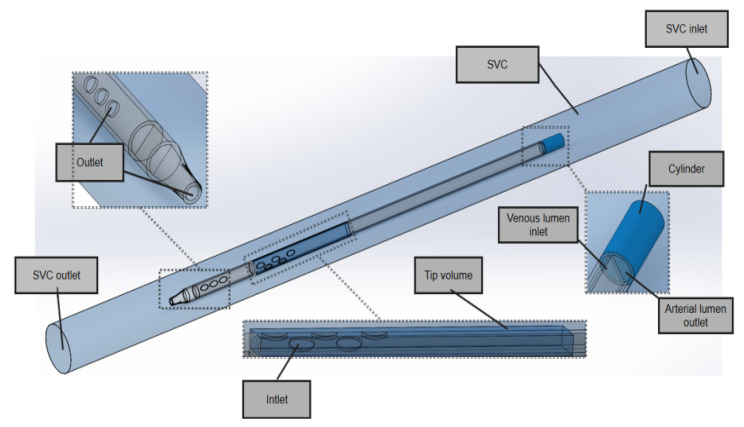
The 3D model of the SVC and the catheter.

**Figure 4 jfb-13-00236-f004:**
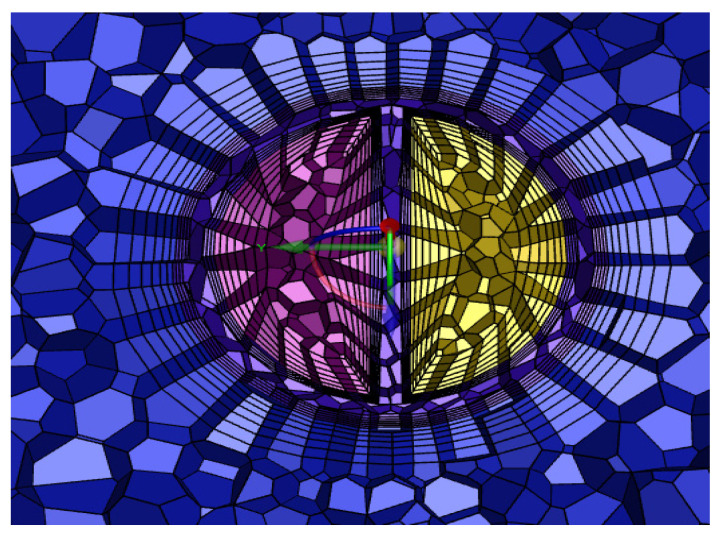
Cross section of grid.

**Figure 5 jfb-13-00236-f005:**
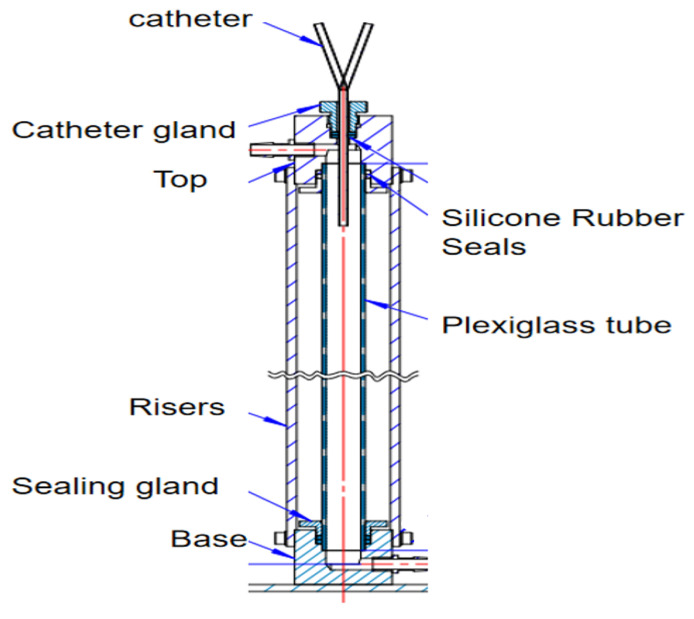
Recirculation test bench for the dye tracing experiment.

**Figure 6 jfb-13-00236-f006:**
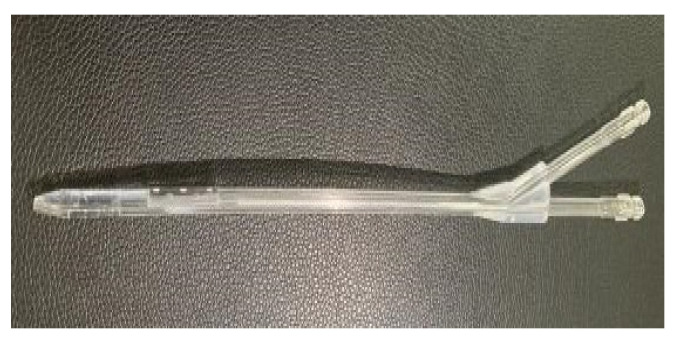
A 3D print of the GDK catheter.

**Figure 7 jfb-13-00236-f007:**
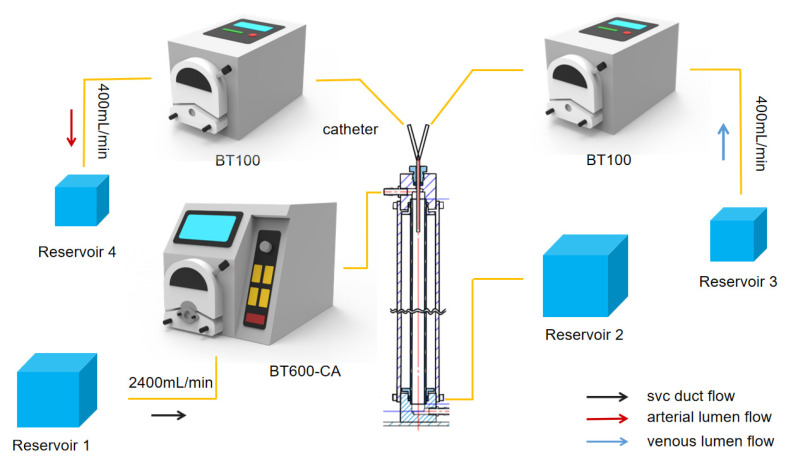
Schematic diagram of the dye tracing device (forward).

**Figure 8 jfb-13-00236-f008:**
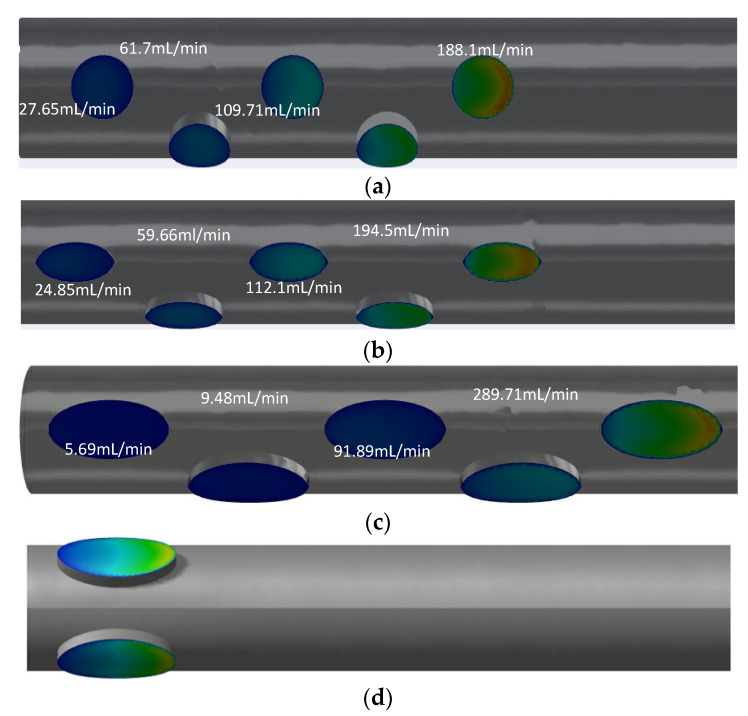
Flow distribution of side holes in the arterial lumen for GDK2 (**a**), GDK3 (**b**), GDK4 (**c**), and GDK5 (**d**).

**Figure 9 jfb-13-00236-f009:**
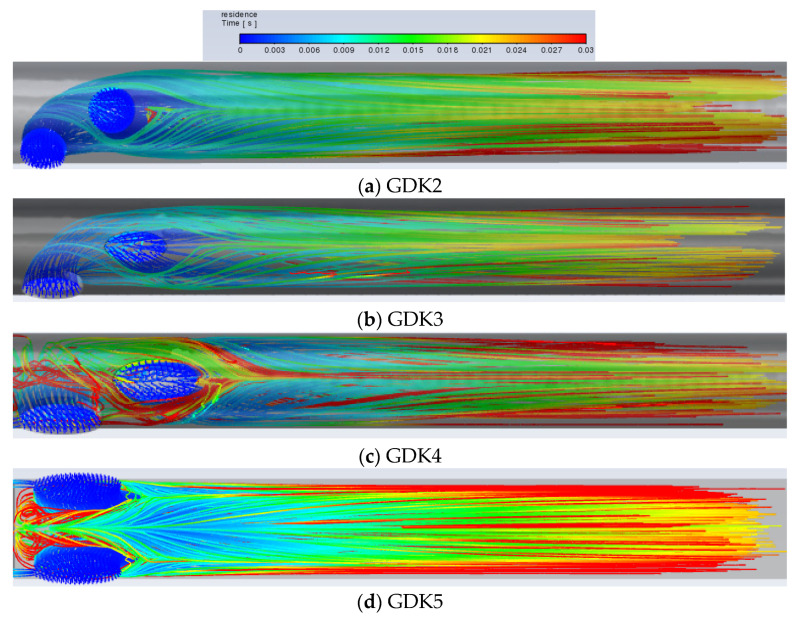
Residence time of platelets in the arterial lumen for GDK2 (**a**), GDK3 (**b**), GDK4 (**c**), and GDK5 (**d**).

**Figure 10 jfb-13-00236-f010:**
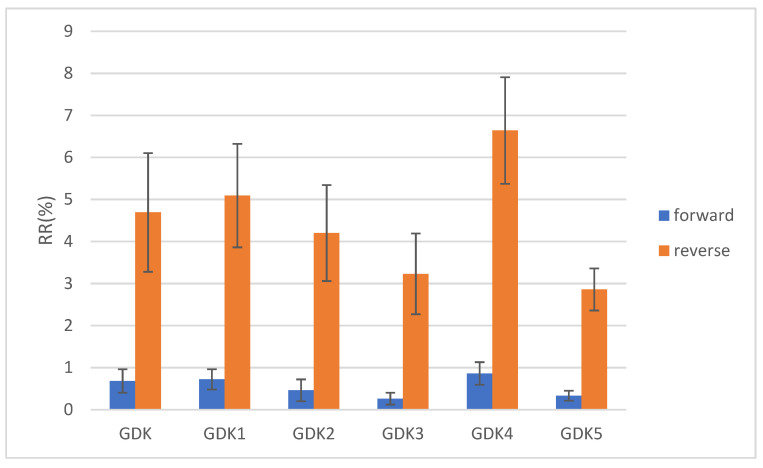
Forward and reverse recirculation rates for the six catheters. The blue columns show RRs (%) for the forward connection and the orange columns show RRs (%) for the reverse connection. The error bars are for RR values (%).

**Table 1 jfb-13-00236-t001:** Various grid numbers and test results for independent tests.

Number of Grids	Maximum Velocity (m/s)	Average Shear Stress (1/s)
94,048	1.65	2.22
220,826	1.83	2.88
320,187	1.78	3.19
420,186	1.78	3.19

**Table 2 jfb-13-00236-t002:** The flow rate through the distal tip and side holes in the venous lumen.

	Q_side-hole1_ (mL/min)	Q_side-hole2_ (mL/min)	Q_side-hole3_ (mL/min)	Q_side_ (mL/min)/ Percentage of Flow Rate through Side Holes	Q_tip_ (mL/min)/ Percentage of Flow Rate through Distal Tip
GDK	89.71	38.74	14.57	143.02 (35.76%)	257.14 (64.28%)
GDK1	89.83	77.41	58.54	225.78 (56.44%)	173.98 (43.49%)
GDK2	74.14	72.34	62.29	208.77 (52.19%)	191.2 (47.8%)
GDK3	135.66	80.57	44.57	260.8 (65.2%)	139.08 (34.77%)

**Table 3 jfb-13-00236-t003:** The maximum flow velocity and average shear stress values at the tip volume for each of the six catheters.

	Max Velocity (m/s)	Average Shear Stress (Pa)	Percentage of Shear Stress > 10 Pa
GDK	5.14	13.6	22.6%
GDK1	3.18	11.0	27.4%
GDK2	1.71	3.43	12.8%
GDK3	1.81	3.15	12.2%
GDK4	1.78	3.19	12.6%
GDK5	1.72	3.16	12.5%

**Table 4 jfb-13-00236-t004:** The values for average RT, PLI, average shear stress, and percentage shear stress > 10 Pa at the tip volume of the six catheters.

	Average RT (s)	PLI	Average Shear Stress (Pa)	Percentage Shear Stress > 10 Pa
GDK	0.008	0.9225	23.107	66.05
GDK1	0.01317	0.173	13.684	45.71
GDK2	0.0304	0.0326	5.867	14.60
GDK3	0.0179	0.0253	5.81	14.3
GDK4	0.0308	0.0174	5.388	13.00
GDK5	0.0409	0.0124	5.386	12.98

## Data Availability

Not applicable.

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
