# Peer review of "Hemodynamic Analysis of the Geometric Features of Side Holes Based on GDK Catheter"

_jfb, 2022, doi:10.3390/jfb13040236_

Round 1
Reviewer 1 Report
The authors Yang Yang et al wanted to know in how far the selection of respective material and the optimization of the side holes of catheters could be of importance for hemodynamic properties and stimulation of coagulation. The paper is technically thorough giving an abundance of detailed information on the topic of the authors’ work.
I have only a few remarks:
(1) Some sort of statement or hypothesis should be given at the end of the introduction: What did the authors test or wanted to know specifically in their project ?
(2) The description of the methods is enriched with a lot of details and technical information. The authors should somewhere give a short overview of this section.
(3) Recirculation test bench: As far as I understand the setup is not used for an in vivo evaluation. The catheters are still not tested in patients. Furthermore, the description of the test bench should be integrated into methods.
(4) The authors should start the discussion with a short comprehensive statement on the major findings, the consequences of these findings and – if possible - the impact that these findings could have in everyday dialysis care.
Reviewer 2 Report
The Methods section does not describe how experiments were done.
How many experiments were performed? Please provide statistical analysis.
Figure legends need more descriptions.
Fig. 10: How was RR(%) calculated?
I presume “4. In-vivo experiment for recirculation” should be “4. In-vitro experiment for recirculation”.
The Abstract section needs to include more specific descriptions of what were found in this study. For example, just by reading the Abstract, readers would not know what “The Lower PLI was seen with the GDK3 and GDK4 catheters.” Since the abstract does not describe what GDK3 and GDK4 are.
Round 2
Reviewer 2 Report
Please provide means +/- SEM values in the results, perform appropriate statistical analysis, and then make conclusions.
Round 3
Reviewer 2 Report
It would be easier for the readers to see if the authors indicate SEM values on the bars of Fig. 9 with appropriate notations of significant differences.
